# The Eye of the Storm: Investigating the Long-Term Cardiovascular Effects of COVID-19 and Variants

**DOI:** 10.3390/cells12172154

**Published:** 2023-08-27

**Authors:** Nandini Vishwakarma, Reshma B. Goud, Myna Prakash Tirupattur, Laxmansa C. Katwa

**Affiliations:** Department of Physiology, Brody School of Medicine at East Carolina University, Greenville, NC 27834, USA; vishwakarman18@students.ecu.edu (N.V.); rbgoud@ncsu.edu (R.B.G.); tirupatturm18@students.ecu.edu (M.P.T.)

**Keywords:** COVID-19, SARS-CoV-2, variants, long COVID, cardiovascular disease, acute cardiac injury

## Abstract

COVID-19 had stormed through the world in early March of 2019, and on 5 May 2023, SARS-CoV-2 was officially declared to no longer be a global health emergency. The rise of new COVID-19 variants *XBB.1.5* and *XBB.1.16*, a product of recombinant variants and sub-strains, has fueled a need for continued surveillance of the pandemic as they have been deemed increasingly infectious. Regardless of the severity of the variant, this has caused an increase in hospitalizations, a strain in resources, and a rise of concern for public health. In addition, there is a growing population of patients experiencing cardiovascular complications as a result of post-acute sequelae of COVID-19. This review aims to focus on what was known about SARS-CoV-2 and its past variants (Alpha, Delta, Omicron) and how the knowledge has grown today with new emerging variants, with an emphasis on cardiovascular complexities. We focus on the possible mechanisms that cause the observations of chronic cardiac conditions seen even after patients have recovered from the infection. Further understanding of these mechanisms will help to close the gap in knowledge on post-acute sequelae of COVID-19 and the differences between the effects of variants.

## 1. Introduction

Cardiovascular disease (CVD) continues to be the leading cause of death globally, affecting more than 19 million people yearly. In the first year of the SARS-CoV-2 pandemic, the number of CVD deaths increased from 874,613 in 2019 to 928,741 in 2020, the highest rise of single-year cases since 2015 [1]. In 2022, Xie et al. found that COVID-19 patients showed elevated risks and experienced increased incidences of CVD in a 12-month period [2]. With an increasing number of patients presenting with adverse cardiac outcomes, such as myocarditis, arrhythmias, and myocardial injury, further investigation into the effects of COVID-19 on cardiac health is needed. Since the emergence of COVID-19 in 2019, the global impact of this viral infection has been extensive, causing widespread morbidity and mortality [3]. COVID-19 can also affect more vulnerable populations, such as pregnant individuals, to a higher degree. Our group recently reviewed how COVID-19 infection can worsen cardiovascular health during and after the gestational period, especially as long-term effects can disproportionately impact vulnerable groups [4]. This complex issue necessitates comprehensive investigation to understand the virus’s influence on molecular, cellular, and organ tissue systems. This information can then in turn be translated to improve both maternal and fetal health. Initially, the focus was primarily on one or two variants; however, subsequent waves and mutations have presented new challenges [5]. From the beginning, COVID-19 has evolved with thousands of variants and mutations. Before, COVID-19 was simply a respiratory infection, but now we know that all organ systems are infected. This review aims to summarize the current understanding of the long-term consequences of COVID-19, specifically in relation to cardiovascular disease (CVD) and various variants.

While the exact mechanism of how COVID-19 affects the heart is not concrete, the following proposed mechanism highlights a direct and indirect pathway taken. The structure of SARS-CoV-2 consists of two spike proteins that bind to Angiotensin I-converting enzyme 2 (ACE2) receptors. Upon entry to the cells, SARS-CoV2 downregulates ACE2 and upregulates TNF-α, an inflammatory cytokine. The downregulation of ACE2 inhibits Angiotensin 1–7 but promotes Angiotensin II (Ang II), which is known for its hypertensive properties [6,7]. This, in turn, promotes TGF-β1, which increases collagen production and ultimately leads to fibrosis and heart failure. Additionally, when SARS-CoV-2 enters the cell, it causes direct damage to the myocardium [6,7]. Different cardiac cells have different interactions with SARS-CoV-2. Endothelial cells that line the vessels of the heart have increased rates of apoptosis due to the virus. This, in turn, causes blood clots to form along with the blockage of arteries. Furthermore, the direct invasion of pericytes, which help regulate endothelial function, can lead to dysfunction and apoptosis as well [8]. Cardiomyocytes affected via the ACE2 receptor and acting through an endosomal protease-dependent pathway increase cytokine production, sarcomere destruction, and apoptosis. This could potentially be extended to endocardial cells as well; however, this area still needs to be investigated [9]. Our lab has previously explored the idea that there are phenotypic changes occurring at a cellular level during the duration of COVID-19 infection. It was found that elevated levels of Angiotensin-II can expedite the phenotypic switch of fibroblasts to myofibroblasts through the TGF-β1 signaling pathway [10]. It is noted that in severe COVID-19 patients, there are increased Ang II levels, which promote the differentiation of fibroblasts to myofibroblasts [10]. However, this is only the tip of the iceberg; the reality is that all organ systems are affected by the infection.

This review seeks to elucidate the prevalence, possible mechanisms, and severity of post-COVID-19 cardiovascular complications. Additionally, we aim to explore the potential pathways by which SARS-CoV-2 and its variants interact with the cardiovascular system. Given the pandemic’s evolving nature and the emergence of new variants, it is critical to continue monitoring and understanding the long-term cardiovascular effects of COVID-19. The findings of this review will highlight the gaps in knowledge and underline the importance of this field of study for potential therapies to mitigate adverse cardiovascular outcomes in COVID-19 survivors.

## 2. Early Discoveries and Variants

Over time, significant strides have been made in understanding SARS-CoV-2. However, there is still more to investigate. This section focuses on the early variants of COVID-19 (Alpha, Delta, and Omicron) and its immediate risks, long-term effects, and cardiovascular complications.

### 2.1. Alpha, Delta, Omicron and Their Immediate Risks

The first highly publicized variant of SARS-CoV-2 was Alpha (*B.1.1.7*). Beginning in the United Kingdom, Alpha rapidly spread across the country and eventually transmitted internationally. On 29 December 2020, it was deemed to be a Variant of Concern (VOC). Alpha ran rampant for 9 months, becoming the dominant variant worldwide. As Alpha began to dwindle down, a new variant came to rise, known as Delta (*B.1.617.2*). Alpha was downgraded to a Variant Being Monitored (VBM) and Delta was declared a VOC on 15 June 2021. Delta’s rise originated in India and rapidly expanded. By the end of 2021, it was dominant in almost every country. The current dominating variant, Omicron (*B.1.1.529*), surfaced in South Africa. It was classified as a VOC on 26 November 2021. To this day, Omicron and its various variants prevail internationally, having been a VOC since as noted in Figure 1 [11,12].

While each variant has mostly similar symptoms to the others, studies show a difference in prevalence. In a recent study observing the difference between Delta and Alpha variants, the Delta variant was more pervasive in headache (75%), rhinorrhea (71%), anosmia/dyssomnia (64%), sneezing (59%), sore throat (56%), and persistent cough (51%) compared to Alpha. Additionally, the odds of five or more symptoms in the first week of infection were higher in Delta compared to Alpha [13]. All Alpha-, Delta-, and Omicron-infected patients exhibited systemic/inflammatory and cardiorespiratory symptoms; however, Alpha patients had a larger expression of symptoms (42.9%), followed by Delta (33%), and then Omicron (20%). With respect to the co-morbidities (diabetes, chronic neurological diseases, neuromuscular disorders, and immunodeficiency), more Omicron patients showed larger percentages in those risk factors. Omicron patients also had a shorter length of stay in the ICU than Alpha- and Delta-infected patients. Although Omicron is considerably less severe than the Delta and Alpha variants [14], it is more transmissible than both [15].

SARS-CoV-2 and its different strains are known for their varied specificity. Some affected individuals may present as asymptomatic, while others suffer dire immediate symptoms [16]. In a recent study, researchers found the key to asymptomatic patients: a mutation in the human leukocyte antigen (HLA). This mutation, *HLA-B*15:01*, still allows for the virus to infect cells; however, it prevents symptoms from arising [17]. Some factors make a person more susceptible to infection, such as age, race, ethnicity, immunocompromised, and past medical conditions [16] Compared to 18–29-year-olds, the hospitalization rate among those aged 50–64 is 3.1 times higher. This rate continues to increase as age advances, with the highest rate observed in the age group of 85 and older, reaching a staggering 15 times higher. When examining the death rate, individuals aged 30 to 39 are 3.5 times more likely to die from COVID-19 compared to 18- to 29-year-olds. As is the case with the relationship between the hospitalization rate and age, the death rate of COVID-19 progressively increases with age. Those who are 85 years old and older had the highest death rate, which was 360 times greater than 18- to 29-year-olds [18]. Similar patterns are found with immunocompromised COVID-19 patients and those with other medical conditions, such as diabetes, asthma, heart conditions, etc. Common immediate risks include but are not limited to loss of taste or smell, fever or chills, difficulty breathing/shortness of breath, fatigue, persistent pain or pressure in the chest, and new confusion [19]. Additionally, acute cardiovascular complications, such as arrhythmia, myocardial injury, acute coronary syndrome, and heart failure, were found to be caused by COVID-19 [20], with myocardial injury and arrhythmia being the most common followed by heart failure, and acute coronary syndrome [7]. In 2020, Zhou et al. found a significant increase in heart failure and acute cardiac injury from surviving patients to non-survivors. Acute cardiac injury jumped from 1% in surviving patients to 59% in non-survivors. Likewise, heart failure rose from 12% to 52%. These adverse cardiac outcomes frequently arise from patients with pneumonia [21]. Evidence shows that acute respiratory infections, such as SARS-CoV-2, can lead to cardiovascular complications [7]. The complexities do not end there. Often, many patients with acute symptoms develop significant long-term cardiac ailments [20].

### 2.2. Long-Term Effects of Alpha, Delta, and Omicron

Previously we established that Omicron is less severe than Alpha and Delta but more transmissible. Therefore, do the Omicron, Delta, and Alpha variants follow the same connection with long-term effects and sequelae as they do with acute symptoms? Hernández-Aceituno et al. found that Omicron patients were less likely to develop long-term sequelae (*p* < 0.001) while Alpha-patients had the highest probability (*p* = 0.016), with Delta-patients falling in the middle. A possible explanation for this finding could be related to Omicron’s lack of severity compared to other variants [14]. Alternatively, perhaps it could be due to the increased amount of vaccination during the Omicron period compared to the original Wuhan SARS-CoV-2 stain period. In a study comparing the fatalities of unvaccinated COVID-19 patients, it was found that Omicron and the Wuhan strain had similar severities. Additionally, the Wuhan strain and Omicron had a lower fatality risk compared to Delta [22]. In terms of vaccine effectiveness, a study depicted that the vaccine was less effective against Omicron compared to Delta [23]. Overall, with vaccination status taken into consideration, studies state that Omicron is less severe than Delta with a decreased risk of hospitalization and death [24,25,26]. The difference in the prevalence of symptoms and long COVID between various variants still needs to be explored. Limited research studies indicate a potential difference; however, not enough is known to draw a conclusion [27,28].

### 2.3. Acute Cardiac Complications

While COVID-19 is known for being a respiratory illness, it is increasingly recognized to profoundly impact the cardiovascular system. Common acute cardiovascular conditions include myocardial injury, myocarditis, pericarditis, and acute coronary syndrome, and they notably arise in patients recovering from pneumonia. Additionally, arrhythmia and heart failure can be considered either acute or long-term depending on the infection time period [20]. These acute complications have the potential to develop into long-term effects, such as sudden cardiac death, on-set hypertension, and myocardial fibrosis.

Myocardial injury is one of the most common acute cardiovascular complications in COVID-19 patients, affecting a quarter of them [29]. Furthermore, it affects patients without a prior cardiovascular disease history. With increased mortality rates and long-term complexities, it is vital to understand how SARS-CoV-2 causes myocardial injury and potential methods to reduce its severity. The critical method in diagnosing myocardial injury is elevated troponin levels, which is defined as “the detection of at least one elevated cardiac troponin value greater than the 99th percentile upper reference limit” [29]. Additional factors indicating myocardial injury are echocardiogram abnormalities in the right ventricle and an ST depression on an electrocardiogram [29]. Another potential cardiac consequence of COVID-19 is the development of blood clots. The virus can induce a prothrombotic state, increasing the risk of clot formation within blood vessels. It has been reported in preliminary studies that in both vaccinated and unvaccinated patients, the incidence of venous thromboembolism was higher among patients infected with the Omicron variant compared to the Alpha and Delta variants [24]. Additionally, the systemic inflammatory response causes the activated macrophages to release an increased number of cytokines, creating a cytokine storm. This causes an increase in adhesion molecules, vascular inflammation, and inflammatory cell infiltration. This mechanism of injury is seen in many other viral infections [29].

Another common cardiac complication observed is myocarditis. The viral infection triggers an immune response in the form of a cytokine storm, and this can directly impact the heart, leading to inflammation and damage of the heart muscle. Biomarkers of cardiac injury, such as troponins and natriuretic peptides, may be elevated in affected individuals [30]. More specifically, myocarditis occurs when the virus infiltrates the mitochondria of cardiomyocytes, causing oxidative stress [31]. Additionally, oxidative stress can occur due to a cytokine storm. In a case study of the first report of myocarditis in an Omicron patient, a 42-year-old male presented with chest pain, inferior wall ST segment elevation myocardial infarction, and elevated troponin levels. However, when a coronary angiography was performed, there was no evidence of coronary artery disease (CAD). After confirming an Omicron variant infection, cardiac magnetic resonance imaging (MRI) was performed depicting a diagnosis of acute myocarditis. In addition, a 60-year-old male with no history of cardiac complications arrived tachycardiac with a prior syncope episode and high troponin levels. He was confirmed for the Omicron variant and, as was the case with the other patient, his physical examination was not out of the ordinary. After CAD was ruled out, a cardiac MRI was performed that confirmed acute myocarditis [32].

### 2.4. Mechanisms of COVID-Induced Cardiac Complications

There are several proposed mechanisms by which COVID-19 induces acute cardiac dysfunction [24,29]. As shown in Figure 2, the COVID-19 virus has been shown to interfere with apoptotic pathways. This, combined with the immune system response, increases apoptosis in vessels and endocardium, and it also heightens the chances of thrombotic disorders [24,25,29]. The risk of thrombotic disorders is also increased by platelet dysfunction upon inflammation, impaired mitochondrial function, and the resultant increase in reactive oxygen species [33].

The direct pathway through which COVID-19 infection might induce long-term cardiac tissue damage is via action on the ACE2 receptor and the endosomal cysteine protease-dependent pathways, and several issues occur in cardiac tissue with relevance to cardiomyocytes. The effects include cytokine production, sarcomere destruction, and cardiomyocyte apoptosis.

Another pathway involves Angiotensin I-converting enzyme 2 (ACE2), a vital receptor in SARS-CoV-2’s entry into a host cell. The spike protein on SARS-CoV-2 targets ACE2, entering the host cell and causing myocardial cell damage/inflammation, downregulation of ACE2, and Angiotensin 1 to 7 production [19,29]. The downregulation of ACE2 then indirectly causes myocardial injury by promoting organ damage, vasoconstriction, microvascular thrombosis, and endothelial dysfunction. The last proposed pathway of myocardial injury is due to a cytokine storm. A systemic inflammatory response is triggered via B and T immune cells causing an upregulation in cardiomyocyte apoptosis, fibrosis, and blood clotting [29]. Additionally, the binding of the virus to ACE2 receptors by the S protein in turn blocks the ACE2 receptors’ activity and causes an increase in angiotensin II (Ang II) and a decrease in Ang-(1-7) [6]. This leads to increased inflammation, fibrosis, oxidative stress, and apoptosis in cardiac tissue. This response was first proposed after clarifying the disruption of the RAAS system due to SARS-CoV-2 infection [34], and observations of increased Ang II in plasma of critical cases of COVID-19 [35]. With this upregulation of prohypertensive peptide Ang II, effects on the cardiovascular system may not appear until later in life, even after other symptoms of COVID-19 have dispersed.

These mechanisms also have a domino effect on endothelial function. With oxidative stress and inflammation, as well as thrombosis and angiogenesis, comes an alteration in the balance of endothelial protective molecules [36]. This results in many acute complications, such as endothelial dysfunction/endothelitis/endotheliopathy [36]. As endothelial function is important to many parts of the body, this mechanism explains why COVID-19 infections often induce multi-organ failure, such as stroke, lung injury, liver and kidney damage, and reproductive system injury. They have the biggest impact on the cardiovascular system, resulting in myocardial injury, peripheral artery disease, and deep vein thrombosis [36].

**Figure 2 cells-12-02154-f002:**
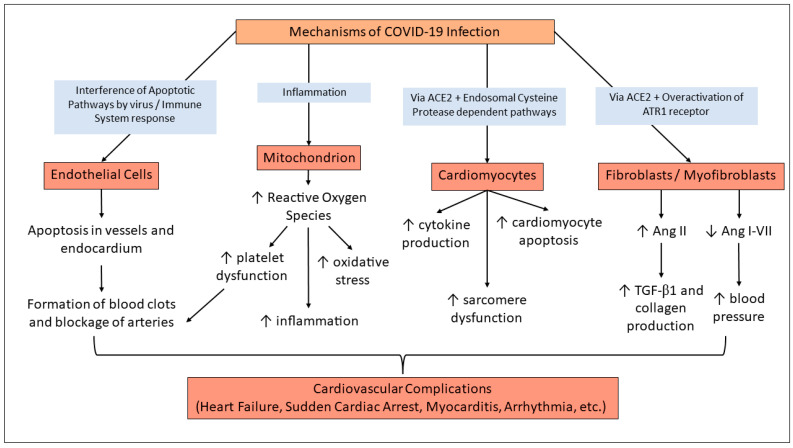
Mechanisms of COVID-Induced Cardiac Complications. SARS-CoV-2 virus infection can affect endothelial cells and cause an increased risk for thrombotic disorders [24,29]. Impaired mitochondrial function has also been demonstrated post-COVID infection and contributes to thrombotic disorders, oxidative stress, and inflammation [33]. Through action on ACE2 receptors, the infection also impacts cardiomyocytes and cardiac tissue by various pathways [29,32]. ACE2 action also produces increased amounts of Ang II, which is a known pathway of ultimate hypertension, cardiac fibrosis, and heart failure [37].

## 3. Emerging Variants and Clinical Observations

### 3.1. Immediate Risks for Rising Variants

On 5 May 2023, SARS-CoV-2 (COVID-19) was officially declared to no longer be a global health emergency [38]. However, a highly infectious and virulent variant of the SARS-CoV-2 virus emerged, posing a significant threat to global health. This section examines the characteristics of this new variant and its association with cardiovascular complications, including sudden cardiac arrest.

New variants of the SARS-CoV-2 virus can impact the transmission dynamic of COVID-19 [24]. Understanding the characteristics of these variants, such as their rate of transmission, severity, and ability to evade immunity, helps public health officials and policymakers to implement appropriate control measures. According to the WHO, as of 27 June 2023, there are two variants of interest (VOI), *XBB.1.5* [39] and *XBB.1.16* [40]. *XBB.1.5* is derived from *XBB*, which originated from the integration of BA.2 variants [38,39]. The WHO has also indicated that there is substantial proof that variant *XBB.1.5* has higher rates of transmission and is more infectious [40]. For example, South Africa has experienced an increase in *XBB.1.5*, rising from 1% in December 2022 to 10% in January 2023 [41], and reaching 83% per the most recent update in May 2023 [42]. On a cellular level, the Omicron subvariant *XBB.1.5* demonstrates a significantly elevated affinity for hACE2 binding in contrast to earlier variants *BQ.1.1* and *XBB*. *XBB.1.5* stands apart from *XBB.1* with a distinctive *Ser486Pro* mutation on the spike protein, showcasing a rare 2-nucleotide substitution that distinguishes it from the original strain [43]. This indicates that this new subvariant has a substantially higher growth advantage over previous variants.

The next current VOI is *XBB.1.16*, which was coined on 17 April 2023 [44]. It is similar to the *XBB.1.5* variant, which has genetic variation on the spike protein, with two supplemental amino acid mutations deemed *E180V* and *K478R* [44]. It is noted that *XBB.1.16* has a high growth advantage because of the increasing trend in positive proportion lineage globally. For instance, from March 2023 to April 2023, the proportion of *XBB.1.16* in COVID-19 cases in India increased from 70.3% to 81.9%, respectively [45]. As this variant is still relatively new, there is controversy in the literature about the clinical severity of this variant. As of 5 June 2023, the WHO’s updated risk assessment deems the level of risk to be low due to no difference in oxygen demand [44]. However, there has been a small percentage increase in hospitalization rates in certain regions of India [45]. A recent study has indicated the inclining trend of the case fatality rate (CFR) in the United States from the oldest to the most recent Omicron subvariants *BA.2*, *BA.4*, and *XBB.1.5*, respectively [46]. This indicates that the most recent Omicron subvariants are increasingly more infectious [46]. Additionally, as of 19 June 2023, there are seven variants under monitoring (VUMs). These include *BA.2.75*, *CH.1.1*, *XBB*, *XBB.1.9.1*, *XBB.1.9*.2, and *XBB.2.3* [47]. This is an indicator that there is potential for growth advantages to increase and for these variants to become VOIs. 

### 3.2. Long-Term Effects of the New Variants

As the COVID-19 pandemic persists, evidence is emerging regarding potential long-term effects correlated with the viral infection. Sudden cardiac arrest (SCA) is a life-threatening condition characterized by the abrupt loss of heart function, leading to a cessation of blood flow and immediate loss of consciousness [48]. In the context of the COVID-19 pandemic, emerging evidence suggests a potential association between COVID-19 and SCA [49]. SCA is gaining attention as a possible long-term consequence of COVID-19 [50]. Multiple mechanisms may contribute to the increased risk of SCA in COVID-19 patients. As discussed earlier, the acute cardiac conditions from COVID-19, such as myocarditis, can disrupt normal cardiac electrical rhythm and function, predisposing individuals to critical arrhythmias [51]. Additionally, as will be further discussed in the following sections, COVID-19 induced systemic inflammation, endothelial dysfunction, and hypercoagulability may promote thrombotic events [52], including coronary artery thrombosis [53], which can trigger SCA. Individuals with a history of severe acute COVID-19 illness or those who experienced myocardial injury during the acute phase may be at a higher risk of SCA in the long term [54].

Several risk factors linked with COVID-19 may contribute to the development of SCA. Advanced age, preexisting cardiovascular disease (such as hypertension, coronary artery disease, or heart failure), and comorbidities such as diabetes and obesity [55] are all known risk factors for severe COVID-19, and they are also linked with an increased risk of SCA [56]. The combination of the SARS-CoV-2 infection and these underlying risk factors may synergistically elevate the risk of SCA. From a study conducted in Italy, one of the first regions to have an outbreak of SARS-CoV-2, there was a notable increase in about 58% in the number of out-of-hospital cardiac arrest cases during the study period in 2020 compared to the same time period in 2019 [57]. It is important to note that these case reports represent individual observations and are not necessarily indicative of a widespread phenomenon. COVID-19 patients who experience SCA often present with a sudden loss of consciousness, absence of a palpable pulse, and respiratory arrest [58]. Prompt recognition and initiation of CPR and defibrillation are essential for successful resuscitation [58]. However, the prognosis of COVID-19 associated SCA remains challenging, with higher rates of mortality compared to non-COVID-19 related SCA [49], highlighting the importance of early detection and rapid response. Future research is needed to deepen our understanding of the underlying mechanisms linking COVID-19 infection and SCA as a long-term effect. Large-scale longitudinal studies are necessary to determine the true incidence, prevalence, and long-term outcomes of COVID-19-associated SCA.

## 4. Long Term Implications

### 4.1. Long COVID

In April 2020, researchers first became aware of post-COVID-19 syndrome when patients noticed residual symptoms [59]. Understanding the long-term effects of COVID-19 can provide a more comprehensive picture of the disease’s impact beyond the acute phase. It will allow healthcare professionals, researchers, policymakers, and the general-public to grasp the full spectrum of health consequences associated with COVID-19, ranging from physical to social to psychological implications. It is also crucial to understand how COVID-19 can impact individuals of all demographics. Adding in the factors of race, sex, and underlying comorbidities adds complexities to the problem at hand and, in turn, makes this public health emergency a complex scientific problem. To find a solution, or at least come close to one, we need to tackle the problem from all angles. Investigating the long-term effects of COVID-19 fuels scientific research and advancements in understanding the disease, and it facilitates ongoing studies to identify underlying mechanisms, risk factors, and potential treatment options. By studying long-term effects, researchers can contribute to the collective knowledge base and facilitate the development of evidence-based guidelines for prevention, treatment, and long-term care.

Post-COVID-19 syndrome, Long COVID, or post-acute sequelae of COVID-19 (PASC) is the persistence of COVID-19 infection/symptoms after 12 weeks involving multi-organ systems [31]. It has been estimated that at least 65 million people present with Long COVID, the highest age group affected being 36- to 50-year-olds. As more studies are being conducted and findings are being uncovered, we have learned that possible signs and symptoms are cardiovascular, thrombotic, and cerebrovascular disease, myalgic encephalomyelitis/chronic fatigue syndrome (ME/CFS), dysautonomia, postural orthostatic tachycardia syndrome (POTS), and type 2 diabetes. These symptoms can carry on for years, and some even for a lifetime (ME/CFS and dysautonomia) [50].

In more recent months, there have been new observations of the manifestation of long-term effects. In a recent study based in the UK and Hong Kong, after 21 days (about 3 weeks), there was a greater risk for many extrapulmonary conditions, such as heart failure, deep vein thrombosis, and atrial fibrillation [60]. As time has passed, there is reason to believe that the potential long-term effects of COVID-19 can become another public health emergency. Another meta-analysis points out the most common long-term effects from 6 months post-infection, and the majority align with the clinical symptoms. These include fatigue, hair loss, headache, and dyspnea [61]. As there are many biological implications for the immune response in the body, studies were performed to indicate whether these immune markers were still present in the body post-COVID-19 infection. A study based in France noted that for those whose symptoms had not fully resolved, there was still no indication of physiological irregularities, such as lymphopenia [62]. Lymphopenia has been found to be an indicator of the severity of disease progression in COVID-19 patients [63]. In terms of cardiovascular disease, evidence depicts cardiovascular outcomes to be significantly higher after COVID-19 compared to pre-exposure, supporting the idea that there is an increase in adverse cardiovascular outcomes due to COVID-19 [2].

### 4.2. Long-Term Cardiovascular Complications

While the immediate impacts of COVID-19 infection are well-documented, the long-term consequences, especially in relation to cardiovascular health, remain a topic of ongoing research. In addition to the various extrapulmonary conditions that can stem from COVID-19, there are many cardiac complications that arise from COVID-19, which is a significant concern. COVID-19 can lead to several cardiovascular manifestations, ranging from mild symptoms to severe adverse effects [64]. As shown in Figure 3, these emerge as risks of conditions that are of higher severity when compared to acute conditions and may lead to further complications.

Some individuals may continue to experience chest pain, heart palpations, and shortness of breath long after the initial infection [65]. These symptoms may be attributed to damage of the heart or ongoing inflammation. It is important to note that the long-term cardiac conditions associated with COVID-19 can affect individuals of all ages, including those with no prior history of cardiovascular disease. Even individuals with mild or asymptomatic cases may develop adverse cardiac outcomes later on. Therefore, it is crucial to monitor the cardiovascular health of all individuals who had been infected with COVID-19, regardless of the severity of their initial symptoms.

Several mechanisms have been proposed to explain the cardiovascular complications connected with long COVID. These include persistent inflammation, endothelial dysfunction, immune dysregulation, and post-viral autoimmune responses. Many studies have found that post-COVID-19 infection, there was a higher risk of individuals developing heart arrhythmias, myocardial infarction, hypertension, and heart failure. The interesting part about this phenomenon is that these conditions may occur even without any previous history of CVD [66]. A recent report indicated that a patient 67 days (about 2 months) after being diagnosed with COVID-19 had abnormal findings when a cardiac MRI was obtained, with no previous underlying conditions [67]. Another factor to consider is that cardiac abnormalities can arise regardless of the age of the patient. It has been found in studies that there is a multiorgan system inflammatory response after COVID 19 in pediatric patients.

### 4.3. Effects of COVID-19 Vaccinations

When comparing the severities of the various variants, vaccination status plays a significant role in symptoms, risk of hospitalization, and death. In a study of unvaccinated COVID-19 patients, myocardial infarction was less present in Omicron patients compared to Alpha and Delta patients. With fully vaccinated patients, Omicron patients also showed lower rates of MI in comparison to Alpha and Delta patients [24]. Additionally, in some cases, vaccines can contribute to cardiovascular or other complications. For example, capillary leak syndrome (CLS) is a rare disease that can be triggered by SARS-CoV-2 and its vaccine. It is hypothesized to be an immune response to an infection via a cytokine storm. CLS occurs when plasma and protein leak out of the capillaries and can eventually cause hypovolemic shock or systemic hypotension. In a case study, a patient diagnosed with COVID-19 and no history of COVID-19 vaccinations suffered from a rapid progression of CLS. The patient, a fit 42-year-old male with no significant medical history, was admitted to the hospital for syncope and hypotension. Other than his blood pressure and increased levels of hemoglobin, all of his other vital signs and blood tests showed no abnormalities. However, over the course of his stay at the hospital, his blood pressure remained low, hemoglobin and hematocrit levels continued to increase, and protein levels decreased. After becoming plethoric and suffering from leg and chest pain, the patient suddenly became unresponsive and died of cardiac arrest [68]. In another case, a patient suffered from CLS after receiving their COVID-19 vaccination (Ad26.COV2.S—Johnson & Johnson vaccine). The patient experienced postvaccination side effects, and 48 h afterward was admitted to the emergency department. Ten hours after admission, the patient died. Throughout his stay, the patient’s blood pressure progressively dropped, and tests showed hemoconcentration and hypoalbuminemia [69]. These case reports, though not common, raise a question to the unknown mechanisms of both COVID infection and vaccination and how they eventually result in such fatal symptoms.

Furthermore, as variants emerge and evolve, reinfection and post-acute sequalae have become common among patients. In a study with over 5 million patients, it was demonstrated that compared to people with no reinfection, people with reinfection showed increased risks of all-cause mortality, hospitalizations, and other adverse outcomes [70]. A specific risk factor was being unvaccinated or being vaccinated before reinfection, and that risk and burden of all-cause mortality increased with the number of reinfections [70]. In terms of cardiovascular complications, people with reinfection exhibited the most risk of cardiovascular disorders after pulmonary sequelae, although the cardiovascular disorders are not specified [70], and there is limited data on cardiac conditions upon reinfection. A recent report by CDC states that reinfection-associated severe outcomes have not been previously characterized, but between September 2021-December 2022, percentages of reinfections have rapidly increased, and the United States is also seeing higher numbers of severe outcomes related to reinfection [71]. Though it may be possible that new variants are evading vaccines, vaccination and antiviral treatment are still recommended to reduce the risk of adverse COVID-19 sequelae.

## 5. Gaps and Future Directions

As previously discussed, there is enough evidence to suggest that Omicron is less severe but more transmissible compared to Delta and Alpha variants. Furthermore, there are many cases of cardiovascular complications linked with COVID-19. However, there is not enough research focusing on the differences in symptoms between each variant or the mechanism of injury in these cardiovascular events. In regard to new variants, due to their recentness, there is also a lack of literature on them as they have not been present for enough time to draw observations. Based on the studies reviewed, the samples utilized to compare different adverse outcomes by variant were solely based on viral genomic sequencing and required the availability of those testing kits, and it therefore covered a limited population.

Most research has been primarily conducted regarding the respiratory system due to COVID-19 being a respiratory illness; however, it has been shown that COVID-19 affects all organ systems, and research should delve more into other areas, such as the cardiovascular system. While the pandemic has been declared over, many people still suffer from the aftermath, and some are living with chronic conditions. This area of research is significant in understanding the effect of post-acute sequelae of COVID-19 and developing therapeutics to help mitigate the adverse effects. Additionally, COVID-19 is relatively new, being present for three years, and with the rise of new variants, it is something that still needs to be monitored and investigated. Further investigation needs to be conducted into the emerging cardiac complications associated with the new COVID-19 variants. Some claims have been made in the media, but there are no studies that highlight this critical concept.

## 6. Conclusions

Looking ahead, several critical areas merit attention for future research on COVID-19 and its variants, with a particular emphasis on adverse cardiovascular outcomes. Investigating the unique cardiovascular impact of each variant, including XBB.1.5 and XBB.1.16, is essential for comprehending the underlying mechanisms of post-acute sequelae of COVID-19. By gaining insights into these variant-specific effects, we can pave the way for targeted therapeutic interventions that lead to improved patient outcomes. To achieve this, conducting longitudinal studies to monitor individuals who have recovered from COVID-19, especially those experiencing emerging cardiovascular complications, becomes paramount. Such studies will provide essential insights into the duration and persistence of post-acute sequelae. Furthermore, it is crucial to identify risk factors associated with prolonged cardiovascular ailments, such as hypertension [7], obesity, and diabetes [31], in people who were infected with COVID-19. This knowledge will play a crucial role in developing early intervention strategies to address these risks effectively.

Additionally, as we have a current understanding of the immunological response that COVID-19 provokes, a Cytokine storm [29], it is worth investigating the relationship between immune responses triggered by different COVID-19 variants. This will deepen our understanding of immunopathogenesis and its contribution to cardiovascular conditions in individuals who were once infected with COVID-19. As more time passes, and additional sequelae present themselves in individuals with long COVID, it is important and beneficial for more research to be conducted on case reports in order to provide clinicians with a more holistic understanding of the treatment and management of these residual symptoms. Continued investigation is necessary to explore novel therapeutic approaches specifically targeting the cardiovascular complexities of COVID-19, especially because some of these individuals did not have underlying comorbidities.

As there are many antiviral therapeutics available for the management of the infection period, it is worth looking into whether these medications can manage other symptoms post-COVID. For example, a study utilizing the Department of Veteran Affairs database noted that the administration of Nirmatrelvir during the infection period correlated with a decreased risk of cardiovascular-specific post-COVID symptoms, including coagulation, deep vein thrombosis, and dysrhythmia [67].

## Figures and Tables

**Figure 1 cells-12-02154-f001:**
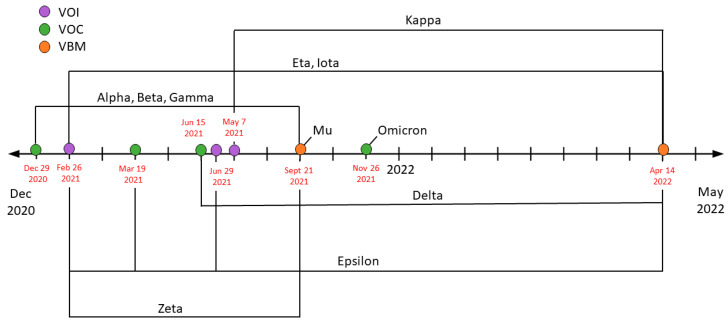
Timeline for COVID-19 variants based on the Center for Disease Control and Protection (CDC). Throughout the pandemic, SARS-CoV-2 has produced multiple variants due to mutations and viral recombination. This timeline depicts when each variant was determined as a Variant of Interest (VOI—purple), a Variant of Concern (VOC—green), or a Variant to Be Monitored (VBM—orange). Additionally, it shows the rise and fall of a multitude of variants; as noted, Omicron remains a VOC.

**Figure 3 cells-12-02154-f003:**
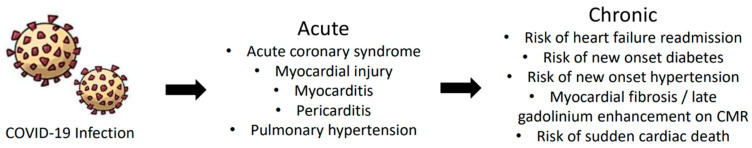
Acute and chronic cardiovascular complications caused by COVID-19. Many times, when patients are presented with acute cardiac symptoms due to COVID-19 infection, it can develop into chronic conditions. The most common acute manifestations are myocardial injury, myocarditis, and acute coronary syndrome. As symptoms worsen and develop, chronic conditions may arise, such as heart failure readmission, sudden cardiac death, myocardial fibrosis, onset hypertension, and diabetes.

## Data Availability

No new data were created or analyzed in this study. Data sharing is not applicable to this article.

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
