# Peer review of "The Eye of the Storm: Investigating the Long-Term Cardiovascular Effects of COVID-19 and Variants"

_cells, 2023, doi:10.3390/cells12172154_

Round 1
Reviewer 1 Report
The comprehensive review summarized the recent progress on the long term cardiovascular complications of COVID-19 and variants. However, the review could be further strengthened to include more discussion on the following issues:
1. In 2.2, the authors mentioned the observation that the long-term complications of Omicron variant appears less frequent than Alpha or Delta ones. The authors raised a possible explanation for this finding could be related to Omicron’s lack of severity compared to other variants. It is not discussed that whether vaccination status contributes to the relatively low long-term complications in Omicron period because the vaccination rate is much higher than Alpha period, and though the Delta causes at least the same severity as Alpha variant, the prevalence of Delta appears lower than Alpha.
2. The review did not address whether there is any evidence demonstrating the effect of vaccination on long-term cardiovascular complications, which is one of the most important questions.
3. It would be better to discuss the possible molecular mechanisms of long-term cardiovascular complications in depth. For instance, for the persistent inflammation in long COVID patients,, what exactly the immune response is impaired resulting in the prolonged inflammation. Any biomarkers could be used for diagnosis?
4. Many molecular mechanisms were proposed for the long-term cardiovascular complications, such as impaired mitochondrial function. Guarnieri JW et al. SCIENCE TRANSLATIONAL MEDICINE 9 Aug 2023 Vol 15, Issue 708. It is better to include the emerging evidence in the discussion.
5. It may be better to break down the long-term cardiovascular complications of COVID-19 to cerebrovascular disorders (TIA, stroke), ischemic heart disease (MI, ACS angina), inflammatory heart disease (myocarditis/pericarditis), Arrhythmia (atrial fibrillation), heart failure, cardiac arrest, thromboembolic disorders (pulmonary embolism) etc, to discuss the prevalence and possible underlying mechanisms, given the difference of prevalence/pathology/mechanisms among these cardiovascular diseases.
6. The emerging variants are capable of evading vaccine/previous infection established immunity. It may be worthwhile to discussing the risk of long-term cardiovascular complications associated with recurrent infections.
Author Response
Reviewer 1
Comment 1: In 2.2, the authors mentioned the observation that the long-term complications of Omicron variant appears less frequent than Alpha or Delta ones. The authors raised a possible explanation for this finding could be related to Omicron’s lack of severity compared to other variants. It is not discussed that whether vaccination status contributes to the relatively low long-term complications in Omicron period because the vaccination rate is much higher than Alpha period, and though the Delta causes at least the same severity as Alpha variant, the prevalence of Delta appears lower than Alpha.
Answer: This is an important idea that we did not consider before. We have included some data on the severity of Omicron and the connection to vaccination rates in lines 151-159 under section 2.2, as well as in lines 430-434.
Comment 2: The review did not address whether there is any evidence demonstrating the effect of vaccination on long-term cardiovascular complications, which is one of the most important questions.
Answer: We have included some of the major cardiovascular effects due to vaccinations in a new section 4.3, lines 434-454.
Comment 3: It would be better to discuss the possible molecular mechanisms of long-term cardiovascular complications in depth. For instance, for the persistent inflammation in long COVID patients,, what exactly the immune response is impaired resulting in the prolonged inflammation. Any biomarkers could be used for diagnosis?
Answer: We have discussed mechanisms more in depth in lines 228-256. In terms of persistent inflammation, we discuss this and immune responses in lines 199-204. Troponin have been identified as the most efficient biomarker for assessing cardiac complications, and this is mentioned in lines 173-177, including ECG methods used for diagnosis. This is also mentioned again in 199-200.
Comment 4: Many molecular mechanisms were proposed for the long-term cardiovascular complications, such as impaired mitochondrial function. Guarnieri JW et al. SCIENCE TRANSLATIONAL MEDICINE 9 Aug 2023 Vol 15, Issue 708. It is better to include the emerging evidence in the discussion.
Answer: Impaired mitochondrial function has been added to Figure 2, as well as discussed in lines 202-204 and 228-256.
Comment 5: It may be better to break down the long-term cardiovascular complications of COVID-19 to cerebrovascular disorders (TIA, stroke), ischemic heart disease (MI, ACS angina), inflammatory heart disease (myocarditis/pericarditis), Arrhythmia (atrial fibrillation), heart failure, cardiac arrest, thromboembolic disorders (pulmonary embolism) etc, to discuss the prevalence and possible underlying mechanisms, given the difference of prevalence/pathology/mechanisms among these cardiovascular diseases.
Answer: We have previously considered several cardiac complications in section 2.4 in this article, along with their mechanisms and doing so again would become repetitive. Additionally, earlier observations have been published in another publication reference #9.
Comment 6: The emerging variants are capable of evading vaccine/previous infection established immunity. It may be worthwhile to discussing the risk of long-term cardiovascular complications associated with recurrent infections.
Answer: A discussion of reinfections and their cardiovascular consequences have been added in lines 464-478.
Additional Changes and Comments:
- With the addition of new references, reference numbers have been adjusted to remain in chronological order.
- Some grammar and diction have been changed in lines 117-119, 173, 202-203, and 491 for better flow and understanding.
- Figure 2 and caption have been changed to accommodate a suggestion by Reviewer 1.
All figures included in this manuscript have been created by the authors.
Reviewer 2 Report
In their manuscript the authors provide some data regarding cardiovasular effects of different variants of SARS-CoV-2. They also describe some molecular mechanisms in association with both acute and long-term cardiovascular effects in patients who suffered from COVID-19.
However, i think that the manuscript is not well structured whereas there is a replication of the discussion of mechanisms in different parts of the manuscript. In addition, many similar reviews and original articles have already been published in the literature.
The authors do not comment enough on the immunological mechanisms and their interrelation with endothelial dysfunction (i.e. CRP). A discussion about capillary leak syndrome should also be added.
I suggest to the authors to provide more data or a table of all studies that have been published so far regarding acute vs chronic cardiovascular complications, so that the reader can better understand the potential mechanisms, predisposing factors and morbidity of such devastating complications of COVID-19.
Author Response
Comment 1: In their manuscript the authors provide some data regarding cardiovasular effects of different variants of SARS-CoV-2. They also describe some molecular mechanisms in association with both acute and long-term cardiovascular effects in patients who suffered from COVID-19.
However, i think that the manuscript is not well structured whereas there is a replication of the discussion of mechanisms in different parts of the manuscript. In addition, many similar reviews and original articles have already been published in the literature.
Answer: Thank you for your review. We understand that it is not well structured and have tried to organize by different complications and their mechanisms and minimized replication throughout the manuscript. We understand that reviews may have been published, but this includes newer data and aims to shed more light on the topic.
Comment 2: The authors do not comment enough on the immunological mechanisms and their interrelation with endothelial dysfunction (i.e. CRP). A discussion about capillary leak syndrome should also be added.
Answer: We have discussed endothelial dysfunction in lines 257-265 and capillary leak syndrome in lines 444-448.
Comment 3: I suggest to the authors to provide more data or a table of all studies that have been published so far regarding acute vs chronic cardiovascular complications, so that the reader can better understand the potential mechanisms, predisposing factors and morbidity of such devastating complications of COVID-19.
Answer: We thank the reviewer for the suggestion, however, a table focused on cardiac complications and studies would be out of our scope, as our focus is looking at the mechanisms that can be investigated as we move forward with research on COVID-19.
Additional Changes and Comments:
- With the addition of new references, reference numbers have been adjusted to remain in chronological order.
- Some grammar and diction have been changed in lines 117-119, 173, 202-203, and 491 for better flow and understanding.
- Figure 2 and caption have been changed to accommodate a suggestion by Reviewer 1.
- All figures included in this manuscript have been created by the authors.
Reviewer 3 Report
Good written synthesis of the available literature upon the subject, presenting evidence in interesting way. I feel the paper could be published as it is.
Author Response
Reviewer 3
Comment 1: Good written synthesis of the available literature upon the subject, presenting evidence in interesting way. I feel the paper could be published as it is.
Answer: Thank you for your review and encouraging comments.
Additional Changes and Comments:
- With the addition of new references, reference numbers have been adjusted to remain in chronological order.
- Some grammar and diction have been changed in lines 117-119, 173, 202-203, and 491 for better flow and understanding.
- Figure 2 and caption have been changed to accommodate a suggestion by Reviewer 1.
- All figures included in this manuscript have been created by the authors.
Round 2
Reviewer 2 Report
The authors have addressed most of the reviewers' comments